# Combining demographic shifts with age-based resistance prevalence to estimate future antimicrobial resistance burden in Europe and implications for targets: A modelling study

Naomi R. Waterlow[1], Clare I. R. Chandler[2,3], Ben S. Cooper[4], Catrin E. Moore[5], Julie V. Robotham[6,7], Benn Sartorius[8], Michael Sharland[5], Gwenan M. Knight[1,2]*

**1** Centre for Mathematical Modelling of Infectious Diseases, Department of Infectious Disease Epidemiology, Faculty of Epidemiology and Population Health, London School of Hygiene and Tropical Medicine, London, United Kingdom, **2** AMR Centre, London School of Hygiene and Tropical Medicine, London, United Kingdom, **3** Department of Global Health and Development, Faculty of Public Health and Policy, London School of Hygiene and Tropical Medicine, London, United Kingdom, **4** Centre for Tropical Medicine and Global Health, Nuffield Department of Medicine, University of Oxford, Oxford, United Kingdom, **5** Institute of Infection and Immunity, City St George's, University of London, London, United Kingdom, **6** AMR and HCAI Division, Chief Medical Advisor's Group, UKHSA, London, United Kingdom, **7** NIHR Health Protection Research Unit in Healthcare Associated Infections and Antimicrobial Resistance, University of Oxford in partnership with the UK Health Security Agency, Oxford, United Kingdom, **8** University of Queensland Centre for Clinical Research (UQCCR), Faculty of Health, Medicine and Behavioural Sciences, University of Queensland, Brisbane, Queensland, Australia

* gwen.knight@lshtm.ac.uk

## Abstract

### Background

Antimicrobial Resistance (AMR) is a global public health crisis. Evaluating intervention impact requires accurate estimates of how the AMR burden will change over time, given likely demographic shifts. This study aimed to provide an estimate of future AMR burden in Europe, investigating resistance variation by age and sex and the impact of interventions to achieve the proposed United Nations (UN) political declaration targets.

### Methods and findings

Using data from 12,807,473 bloodstream infection (BSI) susceptibility tests from routine surveillance in Europe, we estimate age- and sex-specific rates of change in BSI incidence for the 8 bacteria included in European Antimicrobial Resistance Surveillance Network (EARS-Net) surveillance over 2015–2019. This was used to project incidence rates by age and sex for 2022–2050 and, with demographic projections, to generate estimates of BSI burden (2022–2050). Two Bayesian hierarchical models were fitted across 38 bacteria-antibiotic combinations to the 2015–2019 resistance proportion of BSI by year and at the country-level with and without age and sex disaggregation. Inputting the incidence estimates into the "agesex" and "base" model,

**Data availability statement:** Patient level data is available upon request from the European Antimicrobial Resistance Surveillance Network (EARS-Net) from the Surveillance System (TESSy) for those who meet the criteria for access to confidential data. https://www.ecdc.europa.eu/en/publications-data/european-surveillance-system-tessy (data.access@ecdc.europa.eu). All other data is taken from open sources (e.g., EUROSTAT, ECDC, and ONS) and is included in the GitHub repository. The code used in the analysis was written in R, is available from Github [https://github.com/NaomiWaterlow/Future_projections_AMR] and archived in Zenodo [https://doi.org/10.5281/zenodo.15789192].

**Funding:** N.R.W. and G.M.K. were supported by the Medical Research Council UK, https://www.ukri.org/opportunity/career-development-award/ (MR/W026643/1). C.E.M. was supported by ADILA (Wellcome Trust Grant number 222051/Z/20/Z). J.V.R. was supported by the NIHR HPRU in Healthcare Associated Infections and Antimicrobial Resistance (NIHR200915), a partnership between the UKHSA and the University of Oxford. C.I.R.C. was supported by a grant from the British Academy: Just Transitions for AMR (GCPS2\100009). The funders had no role in study design, data collection and analysis, decision to publish or preparation of the manuscript.

**Competing interests:** The authors have declared that no competing interests exist.

**Abbreviations:** AMR, antimicrobial resistance; AST, antimicrobial susceptibility testing; BSI, bloodstream infection; CFRs, case fatality rates; EARS-Net, European Antimicrobial Resistance Surveillance Network; ECDC, European Centre for Disease Control; IPC, infection prevention control; MRSA, methicillin resistance in Staphylococcus aureus; UN, United Nations; UNGA, United Nations General Assembly.

respectively, we sampled 1,000 model estimates of resistant BSI burden by age, sex, and country to determine the importance of age and sex disaggregation. We explored Intervention scenarios consisting of a 1, 5, or 20 per 100,000 per year reduction in infection incidence rate of change or 5 per 100,000 per year reduction in those older than 64 years.

Overall, in Europe, BSI incidence rates are predicted to increase more in men than women across 6 of the 8 bacteria (*Pseudomonas aeruginosa* and *Enterococcus faecium* were the exception) and are projected to increase more dramatically in older age groups (74+ years) but stabilise or decline in younger age groups. We project huge country-level variation in resistance burden to 2050, with opposing trends in different countries for the same bacteria-antibiotic combinations (e.g., aminoglycoside-resistant *Acinetobacter spp.* ranged from a relative difference of 0.34 to 15.38 by 2030).

Not accounting for age and sex results in differing resistance burden projections, with 47% of bacteria-antibiotic combinations estimated to have fewer resistant BSIs by 2030 compared to a model with age and sex. Not including age or sex resistance patterns results in fewer male cases for 76% (29/38) of the combinations compared to 11% (4/38) for women. We also saw age-based associations in projections with bigger differences at older ages.

Achieving a 10% reduction in resistant BSI incidence by 2030 (equivalent to the UN 10% mortality target) was possible only for 68.4% (26/38) of bacteria-antibiotic combinations even with large reductions in BSI incidence rate of change of −20 per 100,000 per year. In some cases, a 10% reduction was followed by a rebound, with the resistant BSI burden exceeding previous levels by 2050. Limitations include reliance on European data and current trends, and the exclusion of factors such as comorbidities or ethnicity.

## Conclusions

Including country-specific, age- and sex-specific resistance levels alongside projected demographic shifts has a large impact on resistant BSI burden projections in Europe to 2030. Reducing this AMR infection burden by 10% will require substantial reductions in infection incidence rates.

---

Author summary
## Why was this study done?

- Infections caused by bacteria that are resistant to antibiotics are a major and growing threat to public health.

- Older adults and men are at higher risk of serious infections, but most estimates of the future burden of drug-resistant infections do not consider how populations are ageing or how infection risk varies by sex.

- Global targets aim to reduce deaths from drug-resistant infections by 10% by 2030 compared with a 2019 baseline, but it is unclear whether this is achievable.

## What did the researchers do and find?

- The researchers analysed data from over 12 million blood tests for bacterial infections in 29 European countries between 2010 and 2019.

- They used these data to predict how rates of serious bacterial infections and the number of drug-resistant blood-stream infections may change in the future (to 2050), considering age, sex, and population changes.

- Their models show that the burden of drug-resistant bloodstream infections varies substantially by country and by bacteria-antibiotic combination.

- Predictions that do not take age and sex into account may miss a large part of the future burden—especially among men and older adults.

- Even with strong public health actions, reducing the rate of bloodstream infection by 10% will be hard to achieve for several bacteria-antibiotic combinations.

## What do these findings mean?

- Future plans to tackle drug-resistant infections must account for context-specific changes in population demographics and differences in infection rate and drug-resistance burden by age and sex.

- Given projected increases in infections across most countries and bacteria–antibiotic combinations, simply achieving a plateau in the burden of antibiotic-resistant infections would represent meaningful progress.

- These predictions are based on current trends and data from Europe, and may not apply in the same way elsewhere. They also do not include all risk factors, such as underlying health conditions or ethnicity, and should be interpreted with caution.

## Introduction

The ageing of our populations has the potential to drive increases in the mortality and morbidity burden of antimicrobial resistance (AMR) due to the increased risk of serious bacterial infection with age [1–4]. The unknown magnitude of this infection burden will hinder efforts to measure the impact of any intervention to control AMR, and yet underscores the urgency of their implementation.

To control the global priority that is AMR, we need to raise awareness among policymakers, develop and drive investment into successful evidence-based interventions [5], as well as improve understanding of what drives the variation in AMR rates [6]. All three of these can be achieved by modelling burden estimates with clear and transparent methods, along with detailed breakdowns of the factors driving change. Moreover, assessing intervention impact requires robust counterfactuals—how would resistance prevalence have changed in the absence of interventions under the current status quo? Each of these actions requires a better understanding and modelling of demographic drivers of AMR.

We have shown in previous work that resistance prevalence in bloodstream infection (BSI) in Europe is strongly determined by the age and sex of the patient [1]. With substantial sub- and national-variation, the consistency and clear shape of

some relationships provide evidence for the inclusion of age and sex in any predictions of future AMR burden. For example, methicillin resistance in *Staphylococcus aureus* (MRSA) prevalence mostly increases with age, with a higher burden in men. For aminopenicillin resistance in *Escherichia coli* there is a common decline with age but still a higher burden in men [1]. Across Europe, multiple single-country- [7–11] or bacterial- [12,13] focused studies suggest that BSI rates are increasing, especially in men, the elderly or both. However, these trends are often reported by age or sex separately. Standardised estimates of rates of infection incidence change for multiple bacteria over all of Europe by age and sex are lacking.

Previous estimates of the future burden of AMR have not provided clarity on the impact of ageing on AMR burden, either by not quantifying the specific impact of demographic shifts or not including the newly described variation in resistance by age and sex [1]. One of the first major AMR burden predictions (the O'Neill report) does not mention differential burden by sex (or gender) or age in the main report [14], although the underlying forecast models do include population size changes. The most recent AMR burden forecasts from the Global Burden of AMR study [6] included age and sex differences in infectious syndromes and case fatality rates, as well as age-disaggregated pathogen distributions with sex disaggregation included where relevant (e.g., for UTIs). However, neither resistance prevalence nor relative risk of mortality by resistance status was age or sex disaggregated, nor were age groupings consistent across the analysis. The complex nature of their methods and lack of interim results make untangling the effects of changing demographics from others difficult.

The need for estimates of future burden levels and trends in AMR has become more acute following the outlining of global targets for AMR control in the 2024 United Nations General Assembly (UNGA) High-Level Meeting on AMR political declaration [15]. For example, a 10% reduction from 2019 levels in AMR mortality globally by 2030 was agreed. To achieve this target a range of interventions need to be enacted, including infection prevention control (IPC) and antibiotic stewardship [16–18]. However, in many settings there is a lack of data and understanding to enable a clear and combined future burden estimate to inform intervention foci and track impact.

In this work, we generate the projections of the future AMR burden in BSI in Europe with and without interventions targeting BSI incidence that account for expected demographic changes. To do this, we generate and combine modelled estimates of trends in BSI incidence with predicted demographic shifts and resistance prevalence differences by age and sex. We focus on Europe due to the wide availability of high-quality data through the European Centre for Disease Control (ECDC). These methods and outputs should support policymakers and individual countries to explore and consider age and sex as key components of baseline trends in AMR and underpin assessment of interventions against AMR.

## Methods

We use three main steps to estimate the future burden of BSIs with AMR to 2030 (aligned to United Nations (UN) target timelines [15]) and to 2050 in Europe (Fig 1): i) we calculate projected BSI incidence rates, ii) combine these with demographic projections to calculate age- and sex-based BSIs, and then iii) use a hierarchical modelling approach to estimate the number of BSIs that are resistant to antibiotics each year. We assess the impact of including age and sex in the models, as well as projecting the impact of interventions to 2050. Our results highlight the dynamics of three key bacteria-antibiotic combinations (*E. coli*-aminopenicillin resistant, *S. aureus*-methicillin resistant, and *Acinetobacter* spp-aminoglycoside resistant), with results for all 38 bacteria-antibiotic combinations presented in S2 and S3 Appendices. We chose these three examples due to these bacteria having a large contributions (>40% [19]) to the etiology of BSIs, a high number of samples, and after exploration of the trends, variation in the country-level burden estimates.

The code used in the analysis was written in R [20], is available from Github [https://github.com/NaomiWaterlow/Future_projections_AMR] and archived in Zenodo [https://doi.org/10.5281/zenodo.15789192].

## Ethics statement

This work to analyse routinely collected data was approved by the London School of Hygiene and Tropical Medicine ethics board (ref 28157).

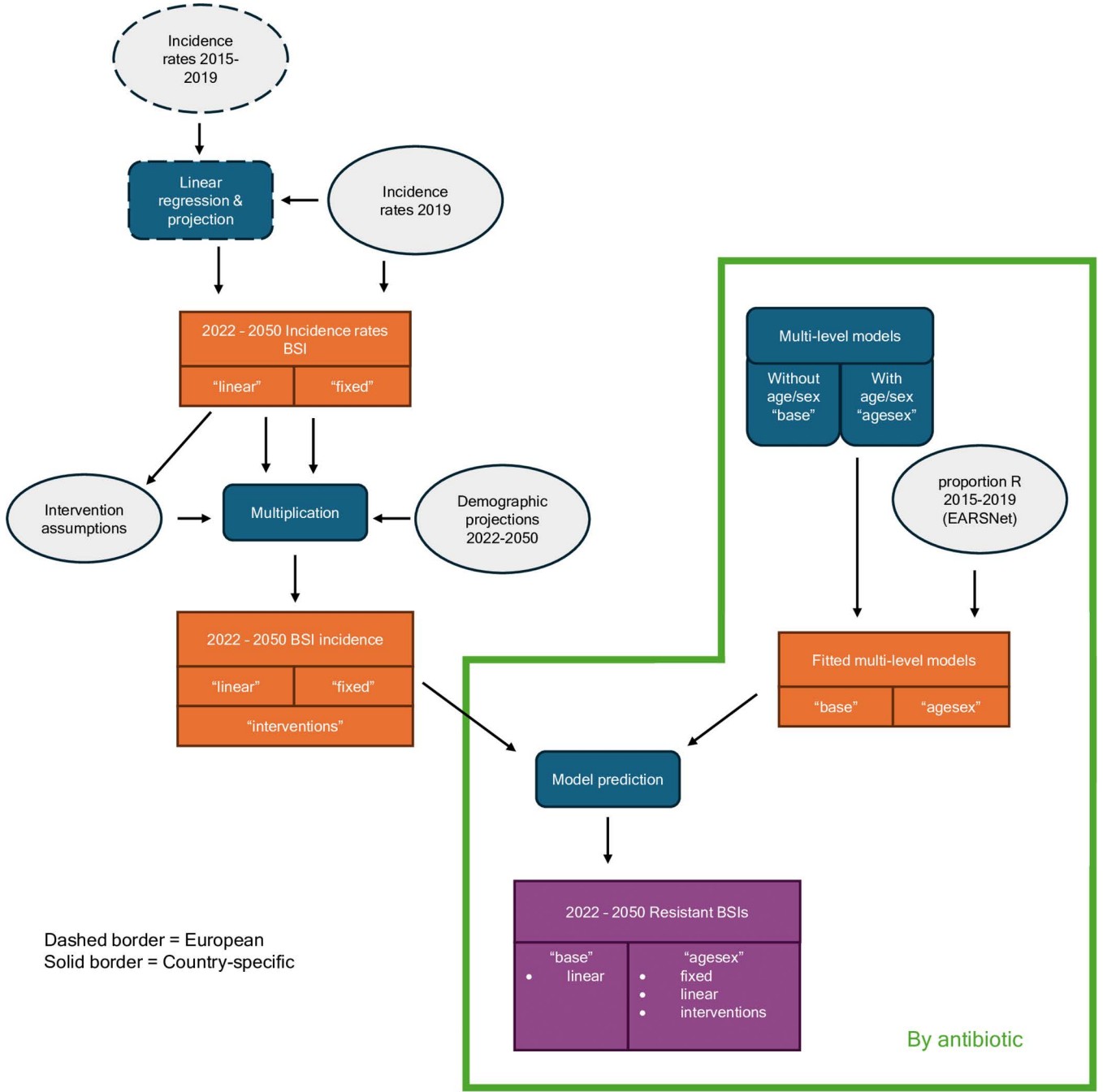

**Fig 1. Overview of methods.** Grey indicates data/inputs, blue indicates modelling steps, orange indicates intermediary results and purple final results. The border indicates whether the data is at a country-specific (solid border) or European (dashed border) level. Items in quotation marks indicate that different models were run with different assumptions. Data and models considered each of the 8 bacteria separately (outside green box) or each antibiotic-bacteria combination (within the green box).

## Incidence and resistance data

We analysed individual patient level data from the European Antimicrobial Resistance Surveillance Network (EARS-Net) Surveillance System (TESSy), provided by Austria, Belgium, Bulgaria, Cyprus, Czechia, Germany, Denmark, Estonia, Greece, Spain, Finland, France, Croatia, Hungary, Ireland, Iceland, Italy, Lithuania, Luxembourg, Latvia, Malta, the Netherlands, Norway, Poland, Portugal, Romania, Sweden, Slovenia, Slovakia, and the United Kingdom and released by ECDC [21]. This surveillance network has collected routine clinical antimicrobial susceptibility testing (AST) results since 1998, alongside some patient data, including age and sex, from EU and EEA countries (we use the term European throughout). The general quality and comparability of the data are evaluated through a standard annual external exercise [22] with the AST results taken from shared protocols [23,24]. The data consists of AST for the first blood and/or cerebro-spinal fluid isolate (<0.7% of this dataset) of every patient with an invasive infection associated with one of the pathogens under surveillance. We exclude individuals aged 0, due to their stark difference in immune dynamics and contact patterns. Data cleaning and analysis of missing data were reported previously [1]. Missingness in the data appeared completely at random with fewer than 3% of isolates missing both age and sex values.

Antibiotic groupings defined as aminopenicillins are ampicillin or amoxicillin, 3G cephalosporins denote cefotaxime, ceftriaxone or ceftazidime, fluoroquinolones cover ciprofloxacin, levofloxacin or ofloxacin, aminoglycosides: gentamicin or tobramycin, macrolides: azithromycin, clarithromycin or erythromycin, penicillins: penicillin or oxacillin, carbapenems: imipenem/meropenem. Please refer to ECDC/WHO report Table 1 for further bacterial-antibiotic specific pairing [25]. We follow the ECDC analysis in taking the recorded data to split by "sex" rather than "gender".

## Incidence rates

Age group- and sex-specific BSI incidence rates between 2015 and 2019 were previously calculated using the EARS-NET data and Cassini and colleagues methodology [1,4]. We projected these forwards using two different assumptions: our "fixed" scenario used the 2019 age- and sex-based incidence rates for all future years, whereas our "linear" scenario projected forward changes in incidence over time. Where no country-specific 2019 estimate was available we use the sex, age-group, and pathogen-specific mean incidence (37% of combinations). For the "linear" scenario, we used a linear regression to estimate the rate of change in incidence from 2015 to 2019 data at the European-level for each 5-year age-band and sex (not taking into account country-level difference due to insufficient data), and then applied this annual rate of change to the country-level age- and sex-specific 2019 incidence rates. This was used to predict BSI incidence estimates for each bacterial species for 2022–2050. We use the "linear" scenario as the base scenario for interventions.

## Age- and sex-specific BSIs

The shift in population age and sex distributions was at the country-level and taken from the EUROSTAT population projections for 2022 onward, and as they are not in the EUROSTAT projections, the ONS predictions for the UK published in 2025 [26,27]. As the baseline, we used the "baseline" EUROSTAT scenario and "Principal" for the UK. As a sensitivity analysis, we used the low mortality ("LMRT") EUROSTAT scenario and the 'old age' projection for the UK. We chose this alternative population projection as the overall population remains stable, but the elderly proportion increases (Fig A1 in S1 Appendix). Incidence rates to 2050 were combined with the projected population demographics.

## Hierarchical modelling of resistance

We developed two hierarchical models which we fit to the individual sample level data from EARS-NET from 2010 to 2019. We did not include data after 2019 due to the impact of the COVID-19 pandemic, as this created temporary changes to many aspects of society that may affect AMR levels [28,29]. This assumes that the trend in resistance by year and the relationship between age, sex, and AMR is unaltered between 2019 and 2022 and beyond.

The first model ("base") included a country-level random intercept, as well as a fixed year and random year-slope effect (Fig 1). The second model ("agesex") included age and sex terms, including a random country slope on age, as informed by our previous modelling work [1]. The models were fit using no-u-turn sampling in R[20] (brms package [30]) and we initially ran the models for 2,000 iterations with package defaults, and extended this to 4,000 or 5,000 if the Rhat for any parameter was larger than 1.01 or the ESS was 100 or lower. In addition, for some models, we changed the controls to avoid divergent transitions. Each model was fit separately for each country and bacteria-antibiotic combination.

In the below model structures, $Y$ is the binomially distributed resistance variable, $n$ is the total number of samples, and $p$ is the probability of resistance. Subscript $c$ represents country-level and $i$ represents grouping level (unique combinations of sex, age in years, and year of sample). A logit link function relates the probability to a linear predictor that includes both fixed and random effects. Fixed effect coefficients are: $\beta 0$ – intercept, $\beta 1$ – year coefficient, $\beta 2$ – age coefficient, $\beta 3$ – age squared coefficient, and $\beta 4$ – sex coefficient. Random effect coefficients are: $b0_c$ – the country-specific deviation from the intercept, $b1_c$ – country-specific effect of year, and $b2_c$ – country-specific effect of age. The random effects follow a multivariate normal distribution.

$$Y_{i,c} \sim Binomial(n_{i,c}, p_{i,c})$$

$$logit(p_{i,c}) = \eta_{i,c}$$

Model 1: "base"

$$\eta_{i,c} = (\beta 0 + b0_c) + (\beta 1 + b1_c)year_{i,c}$$

Model 2: "agesex"

$$\eta_{i,c} = (\beta 0 + b0_c) + (\beta 1 + b1_c)year_{i,c} + (\beta 2 + b2_c)age_{i,c} + (\beta 3)age_{i,c}^2 + (\beta 4)sex_{i,c}$$

To project forward, we inputted the previously calculated 2022–2050 age-sex-country-year-bacteria infection numbers ($n_{i,c}$) into each hierarchical model and then we sampled from each model 1,000 times, to get 1,000 samples of the number of resistant infections by age, sex, year, country, bacteria, and antibiotic.

## Model validation

To assess the validity of the model, we compared the projected number of resistant infections in 2023, with the published ECDC data in 2023, using our "linear" incidence and "agesex" resistance model scenario. In order to compare as close to like-for-like as possible, we multiplied the 2023 ECDC-reported data by the country-specific coverage estimates (already in our incidence estimates) published in the 2023 ECDC report [31]. As the publicly available 2023 data are not available by age or sex, we instead compared the total number of resistant infections by country.

## Interventions

To explore the effect sizes needed to achieve global targets, we estimated the number of resistant infections over time under 3 intervention scenarios targeting rates of BSI incidence. These assumed that the annual fixed rate of change of bloodstream incidence would be reduced by 1, 5, or 20 infections per 100,000 in 2027. Interventions could potentially be age-based due to differing risk factors, hence a fourth scenario assumed the rate of change decreased by 5 infections per 100,000 in only those aged 65 and over (chosen as a common age for determining older individuals for surveillance [31] and implementation of public health policies [32]).

## Results

### Incidence rates to 2030

Assuming no change in BSIs incidence ("fixed", i.e., demographic shifts only), the number of BSI is projected to increase across all pathogens, ranging from a 6.7% (*Acinetobacter spp.*, females) to 13.1% (*E. coli*, males) increase between 2022 and 2030 (Fig 2A). This compares to a range of 22.2% (*S. pneumoniae*, females) to 61.5% (*Klebsiella pneumoniae*, males) increase if we project forward using past country-level incidence trends (i.e., demographic shifts and incidence rate shifts, "linear"). Under this projection ("linear"), BSI incidence also increased more in men than women across 6/8 bacteria by 2030: for example, in *Acinetobacter spp.* the female to male ratio is 0.66 in 2030 versus 0.69 in 2022. *P. aeruginosa* and *E. faecium* were the exception, with a female-to-male ratio of 0.56/0.65 in both years, respectively. BSI incidence does not appear to increase significantly across all age groups (Fig 2B). Age groups 1–14 year olds, 30–44 year olds, and 45–59 year olds show a declining trend across all pathogens in the "fixed" scenario, and 1–14 year olds for *Acinetobacter spp.*, *Enterococcus faecalis*, and *P. aeruginosa* showing a declining trend in the "linear" incidence scenario. The age group with the fastest increasing rate across all scenarios and bacteria is the 90+ age group.

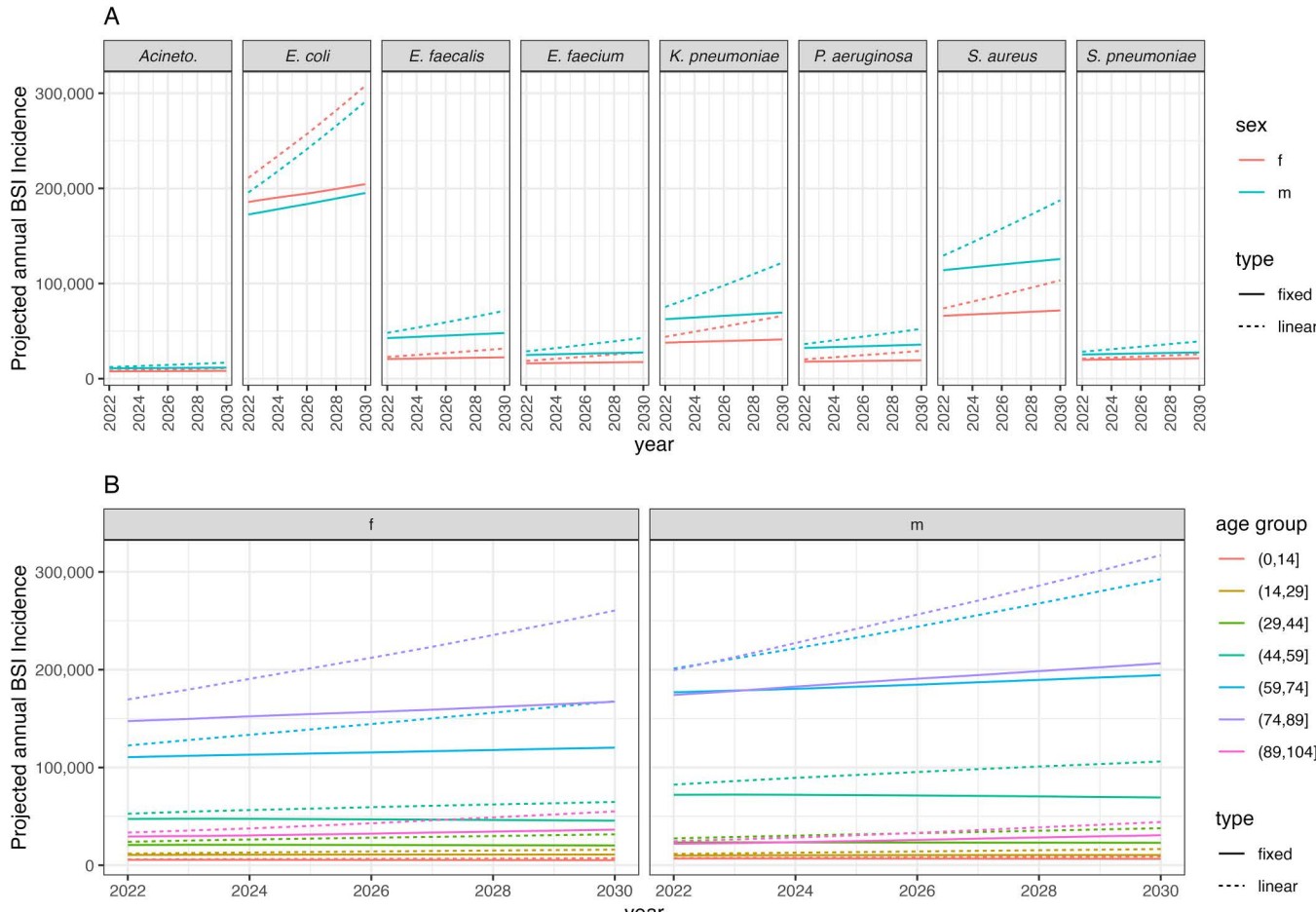

**Fig 2. Incidence of BSIs to 2030, summed across countries (Europe) under different incidence change assumptions: "type" refers to the BSI incidence assumption either "fixed" at 2019 levels or with a "linear" country-level trend. (A)** By bacterial species (panel) and sex (colour). Acineto. stands for *Acinetobacter* spp. **(B)** By 15-year age groups (colour), summed across all 8 bacterial species. Acronyms are female (f) and male (m).

## Trends in resistant BSI

Our modelled number of resistant BSIs was often similar to the best available 2023 data (Figs A1–A12 in S2 Appendix). The data for Malta and *S. pneumoniae* were hardest to match. Model predicted cases of resistant BSI increase to 2050 across 89.0% of bacteria-antibiotic-country combinations, with all bacteria-antibiotic combinations increasing in the majority of countries (Fig 3). The combinations with the lowest number of countries having an increase are both in *E. faecalis* (52.1% and 60.0% for high-level aminoglycoside and aminopenicillin resistance, respectively, Fig 3C). All countries also saw an increase across the majority of bacteria-antibiotic combinations that we had data for, with France (FR) having the lowest percentage of bacteria-antibiotic combinations increasing (70.0%) closely followed by Portugal (PT), and Germany (DE) with 71.0% each. However, despite this, we see large variability in country-level trends, even within bacteria-antibiotic combinations (Fig 3B). For our three key antibiotic-bacteria combinations, the relative difference in resistance burden by 2030, and onwards to 2050, was large. There were opposing trends ranging from 0.34 to 15.38 for *Acinetobacter* spp,-aminoglycoside resistant and 0.33 to 4.91 for MRSA, but consistent increases ranging over 1.3–7.18 for *E. coli*-aminopenicillin resistant (Fig 3B).

## The importance of age and sex

Not including the previously observed trends in resistance by age and sex leads to substantial differences in the projected number of resistant BSI cases in 2030—in particular a variable age and sex breakdown of the cases. Not including age and sex resistance patterns results in fewer estimated resistant BSI cases in 47% (18/38) of bacteria-antibiotic combinations, although these estimates have a high-level of uncertainty (see Table A1 in S1 Appendix). For 76% (29/38) of the combinations, not including age/sex resistance patterns resulted in fewer male cases, compared to 11% (4/38) for women.

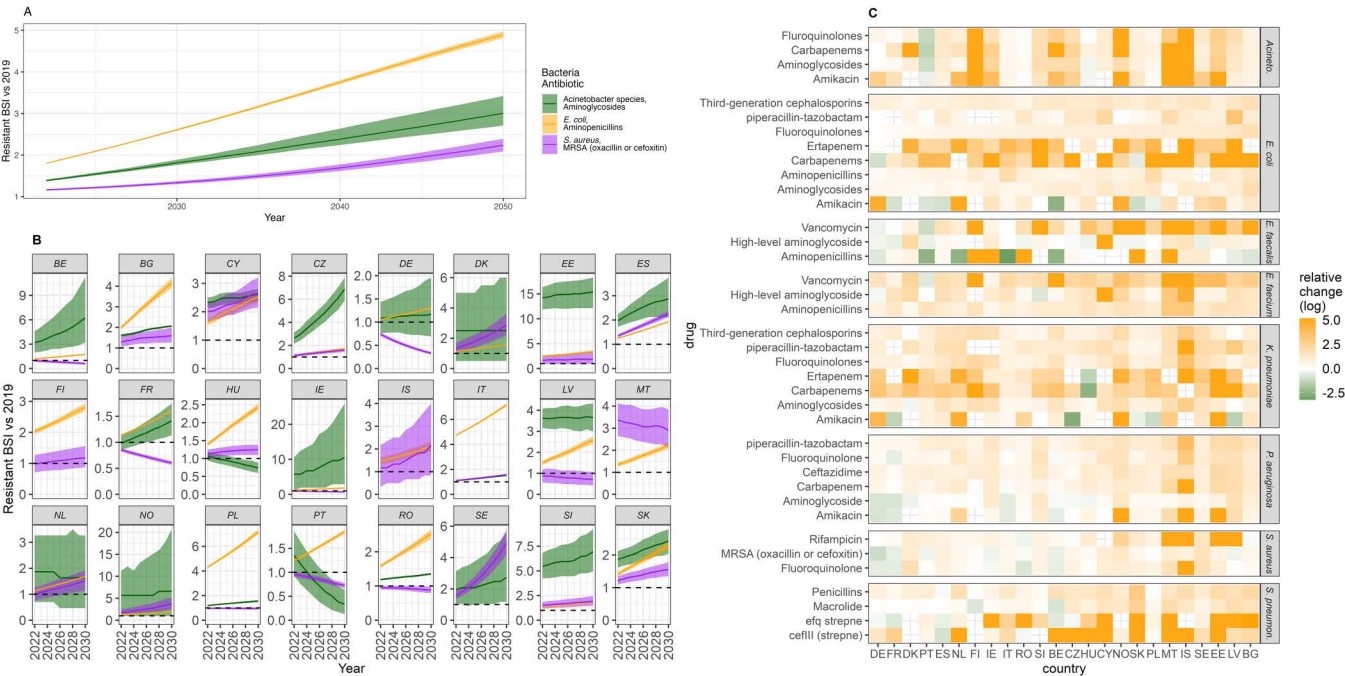

**Fig 3. Projected resistant BSI incidence relative to 2019, using the "agesex" model for resistance levels with "linear" infection incidence trend. (A)** Overall (Europe) and **(B)** country-specific incidence relative changes for three key bacteria-antibiotic combinations (colour). The line depicts the median and the ribbon the 95% quantiles. Only countries with 2019 incidence estimates were included. **(C)** Log of the relative change between 2019 and 2030 by country and bacteria-antibiotic combination.

PLOS Medicine

The age groups across bacteria-antibiotic combinations most affected by demographic assumptions are 71- to 80-year-olds (for females) and 51- to 60-year-olds (for males). For the three example bacteria-antibiotic combinations highlighted (Fig 4), compared to a model with age and sex, there was no strong trend for *Acinetobacter*-aminoglycosides, fewer cases estimated in older men versus older women for *E. coli*-aminopenicillins and more cases estimated in under 70-year-olds and fewer cases estimated for 70+-year-olds for MRSA.

### Intervention impact

Interventions to reduce the rate of change of BSI incidence will have a varying impact by bacteria (Fig 5 and S3 Appendix). Achieving the 'minus 1 per 100,000 change in the rate of increase' intervention only results in 2.6% (1/38) bacteria-antibiotic combinations achieving a 10% reduction in resistant infections by 2030 (when considering the median), compared to 5.36% (2/38) in the "minus 5 per 100,000 change in the rate of increase in ages 65+", 39.4% (15/38) for the

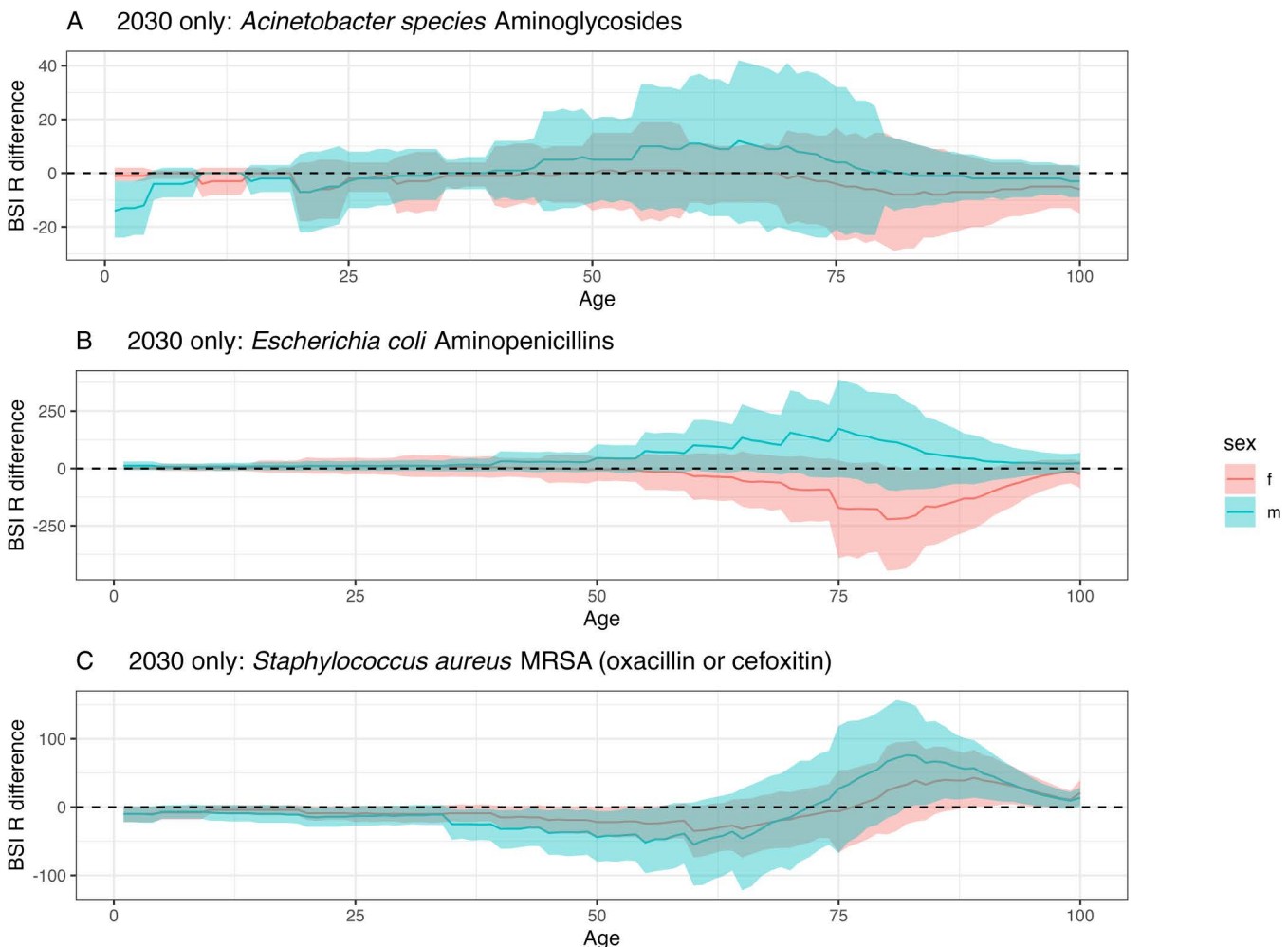

**Fig 4. Difference between 'agesex' and 'base' model projections in 2030.** A positive difference indicates that using the 'base' model estimates fewer resistant BSIs compared to the 'agesex' model. The dashed line at 0 indicates no difference. **(A)**, **(B)**, and **(C)** show the three example bacteria, respectively. The line depicts the median and the ribbon the 95% quantiles. Acronyms are female (f) and male (m). Figures for all bacteria-antibiotic combinations and by country are provided in S2 Appendix.

"minus 5 per 100,000 change in the rate of increase" intervention, and 68.4% (26/38) in the "minus 20 per 100,000 change in the rate of increase" intervention. If considering the upper bound on the projections, the 10% reduction in resistant infections by 2030 is only achieved for the minus 5 and 20 per 100,000 change in the rate of increase interventions (13/38 and 25/38, respectively). Notably, for both *Acinetobacter*-aminoglycosides and MRSA, whilst the 90% reduction in infections was reached with some interventions later than 2030, this threshold was only attained temporarily, and a rebound to increasing numbers of resistance BSI was suggested by the model (Fig 5).

## Discussion

Across eight key bacterial species in Europe, our study projects a future greater increase in BSI incidence among men than women. The most dramatic rises are expected among older adults (aged 74+), whereas incidence rates are predicted to stabilise or decline in younger populations. We identify substantial country-level variation in AMR burden projections to 2050 and show that failure to account for age and sex in resistance burden estimations may result in substantial over- or under-estimation of resistant cases, particularly in older populations, with varying effects by sex. We show that even with very successful public health interventions 32% (12/38) of the included bacteria-antibiotic combinations would

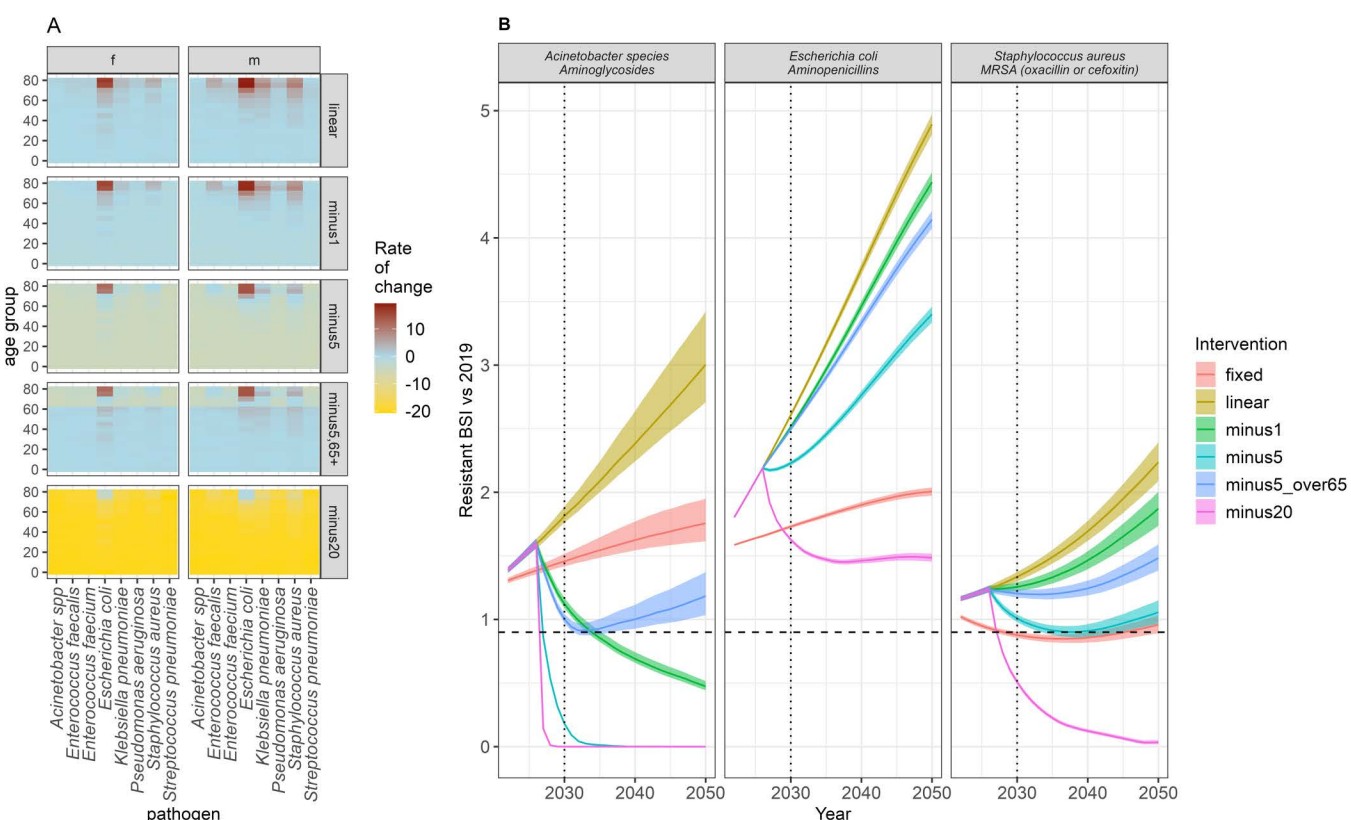

**Fig 5. Intervention impact on BSI rate of change and total number of resistant BSI over time. (A)** The assumed and fixed from 2027 annual rate of change of BSI incidence by age, sex, and pathogen in the baseline (linear) and each of the intervention scenarios. The interventions reduce the annual rate of change of BSI incidence by minus 1/5/20 per 100,000 in all ages or in only those aged 65+ ("minus5_over65") from 2027. This is used to generate the different trends in BSI infection incidence. **(B)** Resistant BSI projections relative to 2019 cases, for each of the intervention scenarios (colour) and the three example bacteria-antibiotic combinations (panels). See S3 Appendix for all combinations. The dashed line is at 0.9 (indicating a 10% relative reduction), and the dotted line at 2030, indicating the UN targets. The line depicts the median and the ribbon the 95% quantiles from 1,000 posterior samples generated using the varying infection incidence calculated with the rate of change in A).

not achieve a 10% reduction in infection burden by 2030, and some that do reach it may subsequently bounce back. Our study employs an open methodology novelly incorporating key demographic factors, offering a new approach to AMR burden estimation.

Our findings align with prior studies highlighting increased BSI incidence among older adults and men [7–10,12]. Whilst EARS-Net was not explicitly designed to monitor infection incidence trends, national surveillance bodies have recognised the importance of supplementing AMR data with infection incidence estimates [33]. With the available information on denominator differences, we propose this here as a new estimate of what is a vitally important indicator (incidence) to calculate future AMR burden. Future work should explore country-level variation as we use European-level age- and sex-based trends in incidence.

We make no assumptions about increasing or declining rates of change in future resistance rates, instead using the large data to project forward existing trends. Moreover, we do this disaggregated by age and sex, accounting for the known differences in levels of resistance by demographic indicators [1] and the potential variation in trend. We did not explicitly account for the impact of COVID-19, though our validation analyses suggest that our projected 2023 resistance incidence levels remain comparable to 2023 data. This matches trends observed in the UK from a different data source, where we see the pre-COVID-19 trend returning in 2023 [34]. However, for several bacteria-antibiotic combinations (such as *S. pneumoniae*), we often had large differences that should be explored more specifically. Hence, our results should be interpreted as relative changes instead of absolute values. These could be applied to post-COVID-19 levels once surveillance and trends have stabilised, or these methods used once we have several years of post-COVID-19 data.

By showing variation at the country-level and reporting relative rates, we hope to emphasise the high levels of uncertainty as well as the need to consider variability by individual bacteria-antibiotic combinations. The GRAM study [6], a major global analysis of AMR burden, does not explicitly incorporate age- and sex-related variations in resistance trends at present based on the limited data available, so this limits direct comparisons. Our results reinforce the importance of stratified analyses to refine AMR burden estimates and intervention strategies. Importantly, we emphasise that the burden and impact of interventions are likely to vary by bacteria-antibiotic combination and that countries need to carefully consider their microbiological composition of AMR burden as well as the demographics of who it affects, to optimally project baseline burden as well as potential intervention impact.

Additionally, we explore the required intervention impact to achieve a 10% reduction in incidence. While not directly correlated with the 10% mortality outcomes set out in the UN political declaration [15], our decision to focus on incidence was due to a lack of comprehensive age- and sex-stratified case fatality rates (CFRs) for these bacterial species. However, given that older adults likely bear the highest mortality burden, we estimated the potential impacts of targeting incidence reduction in individuals aged 65+ only, finding that this often had little impact on AMR infection burden. This was despite assuming a sustained intervention effect on the rate of change of incidence, which would be challenging to implement. Assessing the intervention combination that could achieve this impact is challenging as prior intervention studies have largely compared pre- and post-intervention average rates, highlighting the need for more nuanced evaluations [35–37].

Long-term projections of AMR even to 2030 are inherently challenging [38]—here we could account for demographic shifts but it is impossible to fully incorporate all other dynamic factors influencing resistance burden. Because of this we have not labelled our projections as 'forecasts'—we would only consider them as such in the case that incidence and resistance trends do not vary with time as well as age-sex resistance patterns remaining constant with little long-term impact of COVID-19 on levels or trends. This constancy is unlikely to occur due to interventions such as improved antibiotic stewardship or infection prevention, as well as potential unanticipated outbreaks of new drug-resistant infections or shifts in populations due to migration, so we consider our results as projections, should those conditions be met, rather than 'forecasts'. This allows us to emphasise the importance of age and sex disaggregation of burden and explore intervention impacts.

Driven by available high-quality, individual-level data on AMR burden, our study is limited to Europe. However, the methods could be applied to any setting with representative data. Few other well-curated databases (e.g., GLASS from WHO [39]) concurrently report estimates by age and sex, nor other potentially key factors such as ethnicity. We exclude any individuals aged 0, due to differences in immune function and small numbers (only 1.1% of total resistant BSIs in 2019 in this data). We also had no information to explore and disaggregate likely variable burden by other key factors such as ethnicity and socioeconomic status. Despite the limitations, our approach provides an essential new estimate of AMR burden trends, emphasising the need to account for demographic variations to improve predictive accuracy. Future work is now needed to understand how incidence burden translates to variable mortality burden as well as morbidity, with sequelae likely impacting the burden by age and sex [40].

Future work is needed to explore our projected variation between countries with hugely varying levels of resistance burden to 2030 and onward in our European setting. Country-specific research is needed to untangle incidence, resistance levels, and demographic contributions to current trends in resistance by each antibiotic-bacteria combination. It is known that there is substantial variation in resistance by country [25] with clear national boundaries in burden and microbiology [41,42], likely driven by healthcare networks and antibiotic usage differences in both humans and animals [43–45]. However, the reasons for these differences are still unclear with efforts underway to explore within- [46] and between-country [47] variation hampered by the lack of reliable data outside of high-income settings.

These findings underscore the need for highly successful impactful interventions to meet global targets on AMR [15]. Age- and sex-stratified data, combined with country-level projections for individual bacteria and antibiotics, provide crucial insights for optimising intervention strategies. Tailored strategies incorporating age- and sex-specific risk assessments may enhance the effectiveness of AMR mitigation efforts especially for mortality. Given the substantial uncertainty in AMR burden projections, even in high-income settings with robust surveillance, continued global monitoring and adaptive policy responses are essential. In particular, surveillance and the analysis of trends must always account for diagnostic coverage. Variations in diagnostic coverage may not only lead to bias in resistance burden (with over sampling of more resistant infections often thought to be likely at lower levels of diagnostic coverage [48]) but potentially an undercounting of BSIs. For example, given a growing tendency to limit medical interventions in some elderly populations (such as those with severe comorbidities needing chronic care) this may lead to an apparent reduction in BSI. Further research is needed to elucidate the underlying drivers of age- and sex-related differences in BSI incidence and AMR burden. Understanding these factors could inform more precise intervention strategies and refine predictive models.

Our study provides the specific advance over previous analysis in incorporating demographic shifts into projections of AMR burden in Europe alongside exploring a range of potential intervention impacts to support policy discussion and decision-making in the light of UN targets. We highlight again that, in Europe, older adults and men are likely to experience a higher BSI burden, emphasising the importance of intervention strategies that consider how this burden is shouldered and how it may be mitigated. Failure to incorporate age and sex in AMR burden assessments may result in significant misestimations, underscoring the necessity of stratified analyses by different bacteria-antibiotic combinations as well as by age and sex, supported by improved data disaggregation. While projections to 2050 are uncertain, immediate action is required to mitigate the anticipated rise in resistance burden. Sustained efforts in surveillance, tailored interventions, and demographic-specific policy measures are crucial to effectively combating AMR and reducing the future burden of bloodstream infections. The key implications of our results are the importance of age and sex disaggregation of AMR burden and the difficulty in achieving the AMR targets under the expected demographic population shifts.

## Supporting information

**S1 Appendix. Fig A1: Population distribution for BSL and LMRT scenarios in 2030 and 2050. Fig A2:** Incidence of BSIs to 2030, summed across countries. A) by bacterial species. Acineto. stands for Acinetobacter. "projection" refers to population projection of either "BSL" or "LMRT" (defined differently for the UK, see Methods) B) by 15-year age-band,

summed across all 8 species. Acronyms are female (f) and male (m). **Table A1:** Summary of difference in numbers of resistant BSI between "base" and "agesex" models. Total.diff = the total difference in resistant BSI numbers in 2030 between models. Female.diff and male.diff are the equivalent summaries split by sex. top.ages.female and top.ages.male are the age groups with the biggest differences between models for females and males respectively.
(PDF)

**S2 Appendix.   Fig A1: Log error of predicted BSIs in 2023 compared to ECDC reported data.** Colour indicates country. Dashed lines indicate −0.7 and 0.7, equating to half and double the actual value. **Fig A2:** Predicted BSIs in 2023 under fixed or linear trends (type, colour) compared to ECDC-reported data (black point). A) *Acinetobacter* spp.|Aminoglycosides B) *Acinetobacter* spp.|Carbapenems. **Fig A3:** Predicted BSIs in 2023 under fixed or linear trends (type, colour) compared to ECDC-reported data (black point). A) *Acinetobacter* spp.|Fluoroquinolones B) *Enterococcus faecalis*|Aminopenicillins. **Fig A4**: Predicted BSIs in 2023 under fixed or linear trends (type, colour) compared to ECDC-reported data (black point). A) *E. faecalis*|High-level gentamicin B) *E. faecalis*|Vancomycin. **Fig A5**: Predicted BSIs in 2023 under fixed or linear trends (type, colour) compared to ECDC-reported data (black point). A) *E. faecium*|Aminopenicillins B) *E. faecium*|High-level gentamicin. **Fig A6**: Predicted BSIs in 2023 under fixed or linear trends (type, colour) compared to ECDC reported data (black point). A) *E. faecium*|Vancomycin B) *Escherichia coli*|Aminoglycosides. **Fig A7**: Predicted BSIs in 2023 under fixed or linear trends (type, colour) compared to ECDC-reported data (black point). A) *E. coli*|Aminopenicillins B) *E. coli*|Carbapenems. **Fig A8**: Predicted BSIs in 2023 under fixed or linear trends (type, colour) compared to ECDC reported data (black point). A) *E. coli*|Fluoroquinolones B) *Klebsiella pneumoniae*|Aminoglycosides. **Fig A9**: Predicted BSIs in 2023 under fixed or linear trends (type, colour) compared to ECDC-reported data (black point). A) *K. pneumoniae*|Carbapenems B) *Pseudomonas aeruginosa*|Aminoglycosides. **Fig A10**: Predicted BSIs in 2023 under fixed or linear trends (type, colour) compared to ECDC-reported data (black point). A) *P. aeruginosa*|Carbapenems B) *P. aeruginosa*|Ceftazidime. **Fig A11**: Predicted BSIs in 2023 under fixed or linear trends (type, colour) compared to ECDC-reported data (black point). A) *P. aeruginosa*|Fluoroquinolones B) *Staphylococcus aureus*|Meticillin (MRSA). **Fig A12**: Predicted BSIs in 2023 under fixed or linear trends (type, colour) compared to ECDC-reported data (black point). A) *S. pneumoniae*|Macrolides B) *S. pneumoniae*|Penicillins. **Figs A13–A50**: Each shows the difference between 'agesex' and 'base' model projections in 2030. A positive difference indicates that using the 'base' model underestimates resistant BSIs compared to the 'agesex' model. Figs A13–A50 show the same results for each antibiotic-bacteria combination: Figs A13–A50: difference between 'agesex' and 'base' model projections in 2030. A positive difference indicates that using the 'base' model underestimates resistant BSIs compared to the 'agesex' model. A) by age and sex. The line depicts the median and the ribbon the 95% quantiles. The dashed line at 0 indicates no difference. B) by country. The solid line indicates no difference, and the dotted/ dashed lines represent a 50 and 100% difference, respectively. The dot depicts the median and the ribbon the 95% quantiles. **A13** – Acinetobacter species Amikacin, **A14** – Acinetobacter species Aminoglycosides, **A15** – Acinetobacter species Carbapenems, **A16** – Acinetobacter species Fluoroquinolones, **A17** – *Enterococcus faecalis* Aminopenicillins, **A18** – *Enterococcus faecalis* High-level aminoglycoside, **A19** – *Enterococcus faecalis* Vancomycin, **A20** – *Enterococcus faecium* Aminopenicillins, **A21** – *Enterococcus faecium* High-level aminoglycoside, **A22** – *Enterococcus faecium* Vancomycin, **A23** – *Escherichia coli* Amikacin, **A24** – *Escherichia coli* Aminoglycosides, **A25** – *Escherichia coli* Aminopenicillins, **A26** – *Escherichia coli* Carbapenems, **A27** – *Escherichia coli* Third-generation cephalosporins, **A28** – *Escherichia coli* Ertapenem, **A29** – *Escherichia coli* Fluoroquinolones, **A30** – *Escherichia coli* Piperacillin–tazobactam, **A31** – *Klebsiella pneumoniae* Amikacin, **A32** – *Klebsiella pneumoniae* Aminoglycosides, **A33** – *Klebsiella pneumoniae* Carbapenems, **A34** – *Klebsiella pneumoniae* Third-generation cephalosporins, **A35** – *Klebsiella pneumoniae* Ertapenem, **A36** – *Klebsiella pneumoniae* Fluoroquinolones, **A37** – *Klebsiella pneumoniae* Piperacillin–tazobactam, **A38** – *Pseudomonas aeruginosa* Amikacin, **A39** – *Pseudomonas aeruginosa* Aminoglycoside, **A40** – *Pseudomonas aeruginosa* Carbapenem, **A41** – *Pseudomonas aeruginosa* Ceftazidime, **A42** – *Pseudomonas aeruginosa* Fluoroquinolone, **A43**

– *Pseudomonas aeruginosa* Piperacillin–tazobactam, **A44** – *Staphylococcus aureus* Fluoroquinolone, **A45** – *Staphylococcus aureus* MRSA (oxacillin or cefoxitin), **A46** – *Staphylococcus aureus* Rifampicin, **A47** – *Streptococcus pneumoniae* cefIII (strepne), **A48** – *Streptococcus pneumoniae* efq strepne, **A49** – *Streptococcus pneumoniae* Macrolide, **A50** – *Streptococcus pneumoniae* Penicillins.
(PDF)

**S3 Appendix. All figures in this appendix show the same results for each different antibiotic-bacteria combinations.** Figs A1–A38: Resistant BSI projections relative to 2019 cases, for each of the intervention scenarios (colour) as in [Fig 5](). The dashed line is at 0.9 (indicating a 10% relative reduction), and the dotted line at 2030, indicating the UN targets. The line depicts the median and the ribbon the 95% quantiles. The interventions reduce the annual rate of change of BSI incidence by minus 1, 5, or 20 per 100,000 in all ages or in only those older than 65 (minus5over65). **A1** – Acinetobacter species Amikacin, **A2** – Acinetobacter species Aminoglycosides, **A3** – Acinetobacter species Carbapenems, **A4** – Acinetobacter species Fluoroquinolones, **A5** – *Enterococcus faecalis* Aminopenicillins, **A6** – *Enterococcus faecalis* High-level aminoglycoside, **A7** – *Enterococcus faecalis* Vancomycin, **A8** – *Enterococcus faecium* Aminopenicillins, **A9** – *Enterococcus faecium* High-level aminoglycoside, **A10** – *Enterococcus faecium* Vancomycin, **A11** – *Escherichia coli* Amikacin, **A12** – *Escherichia coli* Aminoglycosides, **A13** – *Escherichia coli* Aminopenicillins, **A14** – *Escherichia coli* Carbapenems, **A15** – *Escherichia coli* Third-generation cephalosporins, **A16** – *Escherichia coli* Ertapenem, **A17** – *Escherichia coli* Fluoroquinolones, **A18** – *Escherichia coli* Piperacillin–tazobactam, **A19** – *Klebsiella pneumoniae* Amikacin, **A20** – *Klebsiella pneumoniae* Aminoglycosides, **A21** – *Klebsiella pneumoniae* Carbapenems, **A22** – *Klebsiella pneumoniae* Third-generation cephalosporins, **A23** – *Klebsiella pneumoniae* Ertapenem, **A24** – *Klebsiella pneumoniae* Fluoroquinolones, **A25** – *Klebsiella pneumoniae* Piperacillin–tazobactam, **A26** – *Pseudomonas aeruginosa* Amikacin, **A27** – *Pseudomonas aeruginosa* Aminoglycoside, **A28** – *Pseudomonas aeruginosa* Carbapenem, **A29** – *Pseudomonas aeruginosa* Ceftazidime, **A30** – *Pseudomonas aeruginosa* Fluoroquinolone, **A31** – *Pseudomonas aeruginosa* Piperacillin–tazobactam, **A32** – *Staphylococcus aureus* Fluoroquinolone, **A33** – *Staphylococcus aureus* MRSA (oxacillin or cefoxitin), **A34** – *Staphylococcus aureus* Rifampicin, **A35** – *Streptococcus pneumoniae* cefIII (strepne), **A36** – *Streptococcus pneumoniae* efq strepne, **A37** – *Streptococcus pneumoniae* Macrolide, **A38** – *Streptococcus pneumoniae* Penicillins.
(PDF)

## Acknowledgments

We are grateful for all the work done by the staff of the participating clinical microbiology laboratories and of the national healthcare services that provided data to EARS-Net.

**Disclaimer:** The views and opinions of the authors expressed herein do not necessarily state or reflect those of the European Centre for Disease Prevention and Control (ECDC). The accuracy of the authors' statistical analysis and the findings they report are not the responsibility of ECDC. ECDC is not responsible for conclusions or opinions drawn from the data provided. ECDC is not responsible for the correctness of the data and for data management, data merging, and data collation after provision of the data. ECDC shall not be held liable for improper or incorrect use of the data.

## Author contributions

**Conceptualisation:** Clare I. R. Chandler, Ben S. Cooper, Catrin E. Moore, Julie V. Robotham, Benn Sartorius, Michael Sharland.

**Data curation:** Naomi R. Waterlow, Gwenan M. Knight.

**Formal analysis:** Naomi R. Waterlow, Gwenan M. Knight.

**Funding acquisition:** Gwenan M. Knight.

**Investigation:** Naomi R. Waterlow, Gwenan M. Knight.

**Methodology:** Naomi R. Waterlow, Gwenan M. Knight.

**Software:** Naomi R. Waterlow, Gwenan M. Knight.

**Supervision:** Gwenan M. Knight.

**Visualisation:** Naomi R. Waterlow, Gwenan M. Knight.

**Writing – original draft:** Naomi R. Waterlow.

**Writing – review & editing:** Clare I. R. Chandler, Ben S. Cooper, Catrin E. Moore, Julie V. Robotham, Benn Sartorius, Michael Sharland, Gwenan M. Knight.

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
