## [Editor Report · Decision Letter 0]

14 Mar 2025

Dear Dr Knight,

Thank you for submitting your manuscript entitled "Combining demographic shifts with age-based resistance prevalence: a modelling estimate of future AMR burden in Europe and implications for targets" for consideration by PLOS Medicine.

Your manuscript has now been evaluated by the PLOS Medicine editorial staff and I am writing to let you know that we would like to send your submission out for external peer review.

Please be advised that we require deposition of new code in a public repository, such as GitHub, and a detailed statement regarding data availability. We also ask that you provide some additional discussion of the results pertaining to country level variation in resistance burden to 2050, given that you call this out in the abstract but do not discuss in any detail in the Results. Pertaining to this issue, please add some thoughts in the Discussion on research needed to understand this variability, and what it means for specific countries (i.e. please discuss the need for country-specific research and what, if any, large-scale efforts are ongoing to address this). Highlighting the specific advance of this study over related publications would be beneficial, and we suggest that you consider tempering your representation of the issues with reference 6 (remove 'opaque' and use a more scientific term in reference to their methods).

Before we can send your manuscript to reviewers, we also need you to complete your submission by providing the metadata that is required for full assessment. To this end, please login to Editorial Manager where you will find the paper in the 'Submissions Needing Revisions' folder on your homepage. Please click 'Revise Submission' from the Action Links and complete all additional questions in the submission questionnaire.

Please re-submit your manuscript within two working days, i.e. by Mar 18 2025 11:59PM.

Kind regards,

Alison Farrell, Ph.D.

Senior Editor

PLOS Medicine

---

## [Decision Letter · Decision Letter 1]

24 Jun 2025

Dear Dr Knight,

Many thanks for submitting your manuscript "Combining demographic shifts with age-based resistance prevalence: a modelling estimate of future AMR burden in Europe and implications for targets" (PMEDICINE-D-25-00946R1) to PLOS Medicine. The paper has been reviewed by subject experts and a statistician; their comments are included below and can also be accessed here: [LINK]

As you will see, the reviewers find the work interesting and potentially important for predicting future AMR trends. After discussing the paper with the editorial team and an academic editor with relevant expertise, I'm pleased to invite you to revise the paper in response to the reviewers' comments. We plan to send the revised paper to some or all of the original reviewers, and we cannot provide any guarantees at this stage regarding publication. We ask that you address the comments of the statistical reviewer (#3), clarify the rationale for the antibiotic-bacteria combinations chosen, and consider the potential effects of changing patterns of diagnosis in the elderly. Please provide an Author Summary with bullet points (the last point should briefly state the study limitations), and please include an email address for the contact for EARS-Net.

We ask that you submit your revision by Jul 15 2025 11:59PM. However, if this deadline is not feasible, please contact me by email, and we can discuss a suitable alternative.

Don't hesitate to contact me directly with any questions (afarrell@plos.org).

Best regards,

Alison

Alison Farrell, Ph.D.

Senior Editor

PLOS Medicine

afarrell@plos.org

Comments from the reviewers:

Reviewer #1: I was invited to review this manuscript, labeled as revision, but I don't recall reviewing an earlier version, and I couldn't find a response to comments made on the initial submission.

The authors have provided a very relevant and important contribution to better predict future trends in AMR infection incidence and burden. It is very obvious from their work that age- and gender incidences should be included. They are - correctly - very careful in their communication about the precision of future incidence rates, and - very wisely - did not include assumptions on attributable mortality or sequelae caused by AMR. As such, this manuscript will set the methodological standard for any future prediction on the burden of AMR.

From a methods point I wondered why the key bug-drug combinations were chosen? Aminopenicillin resistance among E. coli is interesting, but these antibiotics have long been replaced as first-line treatment of E. coli infections (bc of AMR and better alternatives). This could be better explained.

Another relevant question is whether we can assume that age in itself does not influence the incidence of BSI. That incidence is associated with age, and a population that gets - on average - older will thus see more episodes of BSI. Yet, BSI relies on a diagnosis made in healthcare settings, and thus a decline in diagnostics applied may reduce the incidence of BSI. At least in some countries (certainly mine) there is a growing tendency to limit medical interventions, including diagnostics, in elderly, especially in subjects with severe comorbidities and needing chronic care. I would not call "less diagnostics" an intervention, but it may well have the same effect on future BSI rates.

Marc Bonten

Reviewer #2: Population aging is an important driver of the burden of antimicrobial resistance (AMR) because of the disproportionate impact on the elderly and hospitalized populations. This paper aims to estimates future AMR burden in Europe in light of demographic shifts using data from 12 million plus bloodstream infection (BSI) susceptibility tests.

BSI incidence rates are predicted to increase more in men than women across 7 of 56 the 8 bacteria (P. aeruginosa was the exception) and are projected to increase more dramatically in 57 older age groups (74+ years) but stabilize or even decline in younger age groups. Overall, this is an excellent paper and will add usefully to the literature.

Reviewer #3: This is a modeling study of the incidence rate of AMR by age and sex until 2025. Overall the analysis is appropriate and builds heavily on prior work by the authors. My minor comments are:

- There are several claims of under-estimation. As the truth is unknown it is hard to call it under-estimation.

- The models use notation that is never defined (e.g., what do the subscripts i and c refer to).

- Is there a reason no interaction between age and sex was included in the models.

- "Data cleaning and analysis of missing data was reported previously" I would include the missing data component here as it is important to the robustness of the findings.

- The limitation section is good and I think appropriately incorporates how uncertain all of this is.

---

* Please upload any figures associated with your paper as individual TIF or EPS files with 300dpi resolution at resubmission; please read our figure guidelines for more information on our requirements: http://journals.plos.org/plosmedicine/s/figures. While revising your submission, please upload your figure files to the PACE digital diagnostic tool, https://pacev2.apexcovantage.com/. PACE helps ensure that figures meet PLOS requirements. To use PACE, you must first register as a user. Then, login and navigate to the UPLOAD tab, where you will find detailed instructions on how to use the tool. If you encounter any issues or have any questions when using PACE, please email us at PLOSMedicine@plos.org.

* Please ensure that the study is reported according to the appropriate guideline, such as TRIPOD if relevant, and include the completed checklist as Supporting Information. When completing the checklist, please use section and paragraph numbers, rather than page numbers. Please add the following statement, or similar, to the Methods: "This study is reported as per [XXXX] guideline (S1 Checklist)."

FIGURES AND TABLES

SUPPLEMENTARY MATERIAL

REFERENCES

* [FOR POPULATION HEALTH/REGISTRY STUDIES] Please ensure that the study is reported according to the RECORD guideline (available from https://www.record-statement.org) and include the completed checklist as Supporting Information. Please add the following statement, or similar, to the Methods: "This study is reported as per the Reporting of Studies Conducted using Observational Routinely-Collected Data (RECORD) guideline (S1 Checklist)." When completing the checklist, please use section and paragraph numbers, rather than page numbers.

* [FOR POPULATION HEALTH ESTIMATES] Please ensure that the study is reported according to the GATHER statement (available from https://www.equator-network.org/reporting-guidelines/gather-statement) and include the completed checklist as Supporting Information. Please add the following statement, or similar, to the Methods: "This study is reported as per the Guidelines for Accurate and Transparent Health Estimates Reporting (GATHER) statement (S1 Checklist)." When completing the checklist, please use section and paragraph numbers, rather than page numbers.

MODELLING STUDIES

The following list is derived from Geoffrey P Garnett, Simon Cousens, Timothy B Hallett, Richard Steketee, Neff Walker. Mathematical models in the evaluation of health programmes. (2011) Lancet DOI:10.1016/S0140-6736(10)61505-X:

* If pertinent, please provide a diagram that shows the model structure, including how the natural history of the disease is represented, the process and determinants of disease acquisition, and how the putative intervention could affect the system.

* Please provide a complete list of model parameters, including clear and precise descriptions of the meaning of each parameter, together with the values or ranges for each, with justification or the primary source cited and important caveats about the use of these values noted.

* Please provide a clear statement about how the model was fitted to the data, including goodness-of-fit measure, the numerical algorithm used, which parameter varied, constraints imposed on parameter values, and starting conditions.

* For uncertainty analyses, please state the sources of uncertainties quantified and not quantified [can include parameter, data, and model structure].

* Please provide sensitivity analyses to identify which parameter values are most important in the model. Uncertainty estimates seek to derive a range of credible results on the basis of an exploration of the range of reasonable parameter values. The choice of method should be presented and justified.

* Please discuss the scientific rationale for the choice of model structure and identify points where this choice could influence conclusions drawn. Please also describe the strength of the scientific basis underlying the key model assumptions.

* For studies that develop a prediction model or evaluate its performance, please ensure that the study is reported according to the TRIPOD statement (https://www.equator-network.org/reporting-guidelines/tripod-statement) and include the completed checklist as Supporting Information. Please add the following statement, or similar, to the Methods: "This study is reported as per the Transparent Reporting of a Multivariable Prediction Model for Individual Prognosis Or Diagnosis (TRIPOD) statement (S1 Checklist)." For studies using machine learning, please use the TRIPOD-AI checklist. When completing the checklist, please use section and paragraph numbers, rather than page numbers.

---

## [Decision Letter · Decision Letter 2]

13 Aug 2025

Dear Dr. Knight,

Thank you very much for re-submitting your manuscript "Combining demographic shifts with age-based resistance prevalence: a modelling estimate of future AMR burden in Europe and implications for targets" (PMEDICINE-D-25-00946R2) for review by PLOS Medicine.

I have discussed the paper with my colleagues and it was also seen again by one of the original reviewers. I am pleased to say that provided the remaining editorial and production issues are dealt with we are planning to accept the paper for publication in the journal.

********

We look forward to receiving the revised manuscript by Aug 20 2025 11:59PM.   

Sincerely,

Alison Farrell, Ph.D.

Senior Editor 

PLOS Medicine

plosmedicine.org

Requests from Editors:

Specific Comments:

Line 44: please qualify bacteria to explain how the 8 were chosen,

Line 47: “to the 2010-2019 resistance proportion” please clarify what is meant.Line 44-45 says 2015-2019.

Line 63 “age or sex” not “age/sex”, or reword (‘age/sex’ indicates they are equivalent)

Line 64: replace “We also often see strong” with “We also saw strong…”

Line 70: awkward phrasing “ bounces back to above previous”. Please revise.

Last sentence of Methods & Findings: Please provide a statement about methodology limitations.

Line 75: why “even”?

Line 128: remove comma after relationships

Line 133 : delete ‘the’ in ‘the men’

Line 134: “However, these reported trends are often by age or sex separately.” Are often what? Verb needed.

Please avoid claims of primacy/qualify, e.g. line 158

The Methods lack a statistics section. Please add.

General comments:

* Please revise and confirm that your title complies with to PLOS Medicine's style. Your title must be nondeclarative and not a question. It should begin with main concept if possible. "Effect of" should be used only if causality can be inferred, i.e., for an RCT. Please place the study design ("A randomized controlled trial," "A retrospective study," "A modelling study," etc.) in the subtitle (ie, after a colon). E.g. “Combining demographic shifts with age-based resistance prevalence to estimate future antimicrobial resistance burden in Europe” A modelling study”

* Please confirm that your abstract complies with our requirements, including format (three sections: Background, Methods and Findings, and Conclusions) and providing all the information relevant to this study type https://journals.plos.org/plosmedicine/s/submission-guidelines#loc-abstract

* Please ensure that the Introduction ends with a clear description of the study question or hypothesis.

* Please ensure that all abbreviations are defined at first use throughout the text. E.g. Spell out United Nations before use of UN in Abstract.

* Please confirm that all numbers presented in the abstract are present and identical to numbers presented in the main manuscript text.

* In the abstract, please include the important dependent variables that are adjusted for in the analyses.

* It appears that one or more study authors is affiliated with one or more of the agencies that funded the study. Thus, the statement “The funders had no role in study design, data collection and analysis, decision to publish, or preparation of the manuscript” does not apply. Please revise the Financial Disclosure accordingly, as in "[Author name] is [author's role] at [funding agency]. The funders had no other role in study design…..”

* The Data Availability Statement (DAS) requires revision:

For the data are not freely available, please describe briefly the ethical, legal, or contractual restriction that prevents you from sharing it.

* Please add the Github link to the statement on code availability in the data availability statement

* Please consider avoiding the use of red and green in order to make your figure more accessible

Based on guidance from Geoffrey P Garnett, Simon Cousens, Timothy B Hallett, Richard Steketee, Neff Walker. Mathematical models in the evaluation of health programmes. (2011) Lancet DOI:10.1016/S0140-6736(10)61505-X, we require the following for modeling studies:

* Please provide a complete list of model parameters, including clear and precise descriptions of the meaning of each parameter, together with the values or ranges for each, with justification or the primary source cited, and important caveats about the use of these values noted.

* Please provide a clear statement about how the model was fitted to the data including where relevant goodness-of-fit measure, the numerical algorithm used, which parameter varied, constraints imposed on parameter values, and starting conditions.

* For uncertainty analyses, please state the sources of uncertainties quantified and not quantified this can include parameter, data, and model structure.

* Please provide sensitivity analyses to identify which parameter values are most important in the model. Uncertainty estimates seek to derive a range of credible results on the basis of an exploration of the range of reasonable parameter values. The choice of method should be presented and justified.

* Please discuss the scientific rationale for this choice of model structure and identify points where this choice could influence conclusions drawn. Please also describe the strength of the scientific basis underlying the key model assumptions.

Comments from Reviewers:

Reviewer #3: All my comments have been addressed. I congratulate the authors on impressive work.

********

---

## [Editor Report · Decision Letter 3]

15 Sep 2025

Dear Dr Knight, 

On behalf of my colleagues and the Academic Editor, Rebecca Grais, I am pleased to inform you that we have agreed to publish your manuscript "Combining demographic shifts with age-based resistance prevalence to estimate future antimicrobial resistance burden in Europe and implications for targets: a modelling study" (PMEDICINE-D-25-00946R3) in PLOS Medicine.

PRESS

Sincerely, 

Alison

Alison Farrell, Ph.D. 

Senior Editor 

PLOS Medicine